# Transcranial Direct Current Stimulation Can Modulate Brain Complexity and Connectivity in Children with Autism Spectrum Disorder: Insights from Entropy Analysis

**DOI:** 10.3390/bioengineering12030283

**Published:** 2025-03-12

**Authors:** Jiannan Kang, Pengfei Hao, Haiyan Gu, Yukun Liu, Xiaoli Li, Xinling Geng

**Affiliations:** 1College of Electronic & Information Engineering, Hebei University, Baoding 071002, China; kangjiannan81@163.com (J.K.); 13102619580@163.com (P.H.); 2Intensive Care Rehabilitation Department, Ningbo Rehabilitation Hospital, Ningbo 315040, China; nbguhaiyan_2008@163.com; 3Nursing Department, Huaxin College, Hebei Geo University, Shijiazhuang 050000, China; liuyukun166@163.com; 4State Key Laboratory of Cognitive Neuroscience and Learning, Beijing Normal University, Beijing 100875, China; 5School of Biomedical Engineering, Capital Medical University, Beijing 100069, China

**Keywords:** autism spectrum disorder, EEG, transcranial direct current stimulation, entropy, brain complexity, effective connectivity

## Abstract

The core characteristics of autism spectrum disorder (ASD) are atypical neurodevelopmental disorders. Transcranial direct current stimulation (tDCS), as a non-invasive brain stimulation technique, has been applied in the treatment of various neurodevelopmental disorders. Entropy analysis methods can quantitatively describe the complexity of EEG signals and information transfer. This study recruited 24 children with ASD and 24 age- and gender-matched typically developing (TD) children, using multiple entropy methods to analyze differences in brain complexity and effective connectivity between the two groups. Furthermore, this study explored the regulatory effect of tDCS on brain complexity and effective connectivity in children with ASD. The results showed that children with ASD had lower brain complexity, with excessive effective connectivity in the δ, θ, and α frequency bands and insufficient effective connectivity in the β frequency band. After tDCS intervention, the brain complexity of children with ASD significantly increased, while effective connectivity in the δ and θ frequency bands significantly decreased. The results from behavioral-scale assessments also indicated positive behavioral changes. These findings suggest that tDCS may improve brain function in children with ASD by regulating brain complexity and effective connectivity, leading to behavioral improvements, and they provide new perspectives and directions for intervention research in ASD.

## 1. Introduction

Autism Spectrum Disorder (ASD) is a heterogeneous neurodevelopmental disorder primarily characterized by social and interaction impairments, as well as restricted, repetitive behaviors and interests. Patients often exhibit abnormal communication and language delays or deficiencies. This condition typically persists throughout the individual’s life, placing a significant burden on both families and society [1]. Accurate assessment and effective intervention are crucial for improving the quality of life of children with ASD.

Electroencephalogram (EEG) is a non-invasive neurophysiological examination method commonly used in medical and research fields [2]. It directly records electrical signals generated by neuronal activity. EEG offers excellent time resolution, low-cost data recording advantages, and can effectively assess the functional state of the brain within certain frequency bands [3]. With the increasing demand for system dynamics analysis, nonlinear techniques have garnered increasing attention in this field. In particular, entropy-based measures, due to their ability to quantitatively describe the complexity and effective connectivity of EEG signals from multiple dimensions, have become an effective complement to traditional methods [4].

In the study of brain functional complexity in neurological disorders, Sample Entropy (SaEn) is the most commonly used entropy measure. Other common entropy measures include Approximate Entropy (ApEn), Shannon entropy, and Multiscale Entropy (MSE) [5]. Zhang et al. calculated the ApEn and SaEn of ASD patients and normal controls and found that the entropy values of ASD patients were significantly lower, indicating reduced brain signal complexity. They also found that the entropy values of specific brain regions were significantly negatively correlated with the total score of the Autism Diagnostic Observation Schedule (ADOS), suggesting that these entropy measures may serve as potential biomarkers to distinguish ASD individuals from typically developing individuals [6]. Additionally, Kang et al. analyzed resting-state EEG in 43 children with ASD, aged 4 to 8 years, and compared various entropy measures, including SaEn. The results revealed significant differences in the performance of these entropy measures across specific age groups and brain regions, indicating that these entropy metrics can capture different dimensions of brain functional complexity [7]. MSE can effectively represent the complexity of EEG signals at different time scales, providing richer dynamic information than traditional entropy measures. In ASD research, an MSE-based study found that the average MSE value of the mild ASD group was significantly higher than that of the severe ASD group, suggesting that MSE can not only sensitively identify the characteristics of brain functional complexity in children with ASD but also reflect the severity of their symptoms [8]. Catarino et al. analyzed the EEG complexity features of adult ASD patients during a visual matching task using MSE. The results showed that the EEG signal complexity in the parietal and occipital regions of the ASD group was significantly lower than that of the control group, with the differences being more pronounced at higher time scales. They suggested that the reduced complexity in these regions may reflect impaired integration of local neural networks, which are involved in the higher-order integration of visual information processing [9].

In the study of brain effective connectivity in neurological disorders, Transfer Entropy (TE) is an information theory-based measure of effective connectivity [10]. It does not rely on specific interaction models, such as Granger causality [11] or dynamic causal modeling [12], and it is inherently nonlinear [13]. Researchers have also developed new estimation methods to analyze time-series signals using TE. Lobier et al. proposed Phase Transfer Entropy (PTE) [14], which quantifies transfer entropy between phase time series of neuronal signals, effectively overcoming linear mixing effects and volume conduction interference. A recent study used normalized phase transfer entropy in the γ band to analyze brain connectivity in 10 children with ASD. The results showed that the ASD group exhibited significantly lower brain activity in the γ band compared to the TD group, indicating lower activity levels and different information flow distributions in the γ-band brain network of children with ASD [15]. However, despite the preliminary progress in PTE research, EEG studies exploring differences in effective connectivity between children with ASD and TD children using PTE remain relatively scarce, warranting further exploration and validation.

Transcranial Direct Current Stimulation (tDCS), as a non-invasive brain stimulation technique, is characterized by high safety, good adjustability, and broad applicability [16], and it had been considered a potential and feasible intervention for ASD [17]. tDCS works by applying a weak direct current to the surface of the scalp, modulating the excitability of cortical neurons and thus influencing the activity of brain neural networks. Hadoush et al. conducted a randomized controlled trial of tDCS in children with ASD and found that the tDCS treatment group showed significant decreases in the total score on the Autism Treatment Evaluation Checklist (ATEC), as well as in the social skills subscore and the behavior, health, and physical condition subscores. No significant changes were observed in the total score or subscores of the placebo control group [18]. Currently, most studies on tDCS intervention for children with ASD focus primarily on behavioral assessments, and research combining EEG signal analysis to examine brain function changes before and after intervention is limited. However, studies have shown that after 10 sessions of tDCS intervention on the left and right cerebellar hemispheres in children with ASD, the average ApEn value in the right frontal cortex significantly increased, suggesting that tDCS may play a potential role in modulating brain complexity in children with ASD [19]. Other studies have used the Maximum Entropy Ratio (MER) method to measure changes in EEG sequence complexity and found that the MER value significantly increased after tDCS intervention in children with ASD [20]. This suggests that tDCS intervention may be a useful tool for the rehabilitation of children with ASD. It is noteworthy that previous studies have focused only on changes in single entropy measures and have not explored the effects of tDCS on brain function in children with ASD from more comprehensive perspectives, such as multi-scale entropy or phase transfer entropy.

In this study, we first used information entropy methods (including SaEn, MSE, and PTE), to assess the differences in brain complexity and effective connectivity between children with ASD and TD children. Based on PTE, we further calculated dPTE to compare the differences in information flow direction between ASD and TD children and analyzed the signal input and output strength at different electrode positions. Next, we designed a randomized controlled trial, dividing children with ASD into a real stimulation group and a sham stimulation group, to explore the potential effects of tDCS intervention on brain complexity and effective connectivity in children with ASD. To assess the intervention effect, we simultaneously used the Autism Behavior Checklist (ABC) to quantify behavioral changes in children with ASD before and after the intervention.

## 2. Materials and Methods

### 2.1. Subjects

In this study, we initially recruited 26 children with ASD, of whom 2 children withdrew from the experiment for personal reasons. Therefore, we analyzed the EEG data of 24 children with ASD (19 males and 5 females; mean ± SD age: 5.62 ± 1.88 years) and 24 age- and gender-matched TD children (19 males and 5 females; mean ± SD age: 5.91 ± 1.63 years). The ASD diagnosis for all the participants was made by a qualified psychiatrist according to the DSM-5 criteria [21]. None of the ASD children had received any form of brain stimulation therapy prior to this study, and none had used antipsychotic or anticonvulsant medications during or prior to the experiment. TD children were recruited from kindergartens or primary schools and were screened to ensure they had no family history of psychiatric disorders, epilepsy, or head injuries. All participants and their legal guardians were thoroughly informed about the experimental procedure, and written informed consent was obtained from the guardians. This study was conducted in accordance with the Declaration of Helsinki and was approved by the Ethics Committee of Ningbo Rehabilitation Hospital (Approval No. 2023006).

### 2.2. tDCS Interventions

The 24 children with ASD who participated in the experiment were randomly assigned to the real stimulation group (*n* = 12) and the sham stimulation group (*n* = 12) through a random draw. The randomization process was carried out by members of the research team. The real stimulation group consisted of 9 males and 3 females, with an average age of 5.67 ± 1.44 years, while the sham stimulation group included 10 males and 2 females, with an average age of 5.33 ± 1.30 years. The real stimulation group received 40 sessions of tDCS intervention (5 sessions per week for 8 weeks). EEG data were collected before and after the 40 sessions of intervention, and changes in scores on the ABC were simultaneously assessed. During the intervention, the anodal electrode was placed on the left dorsolateral prefrontal cortex (DLPFC), and the cathodal electrode was placed on the right supraorbital region. During the experiment, two round sponge electrodes were soaked in saline to ensure that the impedance remained below 20 KΩ at all times. For the real stimulation group, the stimulation current was set to 1 mA. During each stimulation session, the current gradually increased from 0 to 1 mA within 30 s, was maintained for 20 min, and then gradually decreased to 0 over 30 s. The electrode placement for the sham stimulation group was the same as for the real stimulation group, but the current was only passed through the participant’s scalp during the first 30 s of stimulation to simulate the sensation of real stimulation, without actual neural stimulation.

### 2.3. Data Acquisition

In this study, EEG signals were collected using an 8-channel Electrical Geodesics Incorporated EEG system. Resting-state EEG data were recorded for all children with their eyes open. The EEG recording setup was based on the 10–20 international electrode system, including channels F3, F4, T3, C3, C4, T4, O1, and O2. During data collection, the impedance of all electrodes was maintained below 20 KΩ, with Cz as the reference electrode, and the sampling rate was set to 1000 Hz. During data collection, the children sat in a comfortable chair and remained relaxed. Under the guidance of a technician, approximately 5 min of EEG data were recorded. After data collection, professional staff guided the children’s guardians to complete the ABC scale. The scale contains 57 items covering five major behavioral domains: sensory, social interaction, body and object use, language, and self-care in social life. Each item is scored based on the frequency and severity of the behavior, using a scale of 1–4, with a total score of 158. After the completion of the 40 interventions, another round of data collection and scale completion was conducted.

### 2.4. Data Preprocessing

Offline data processing was performed using MATLAB R2016a and EEGlab v2022.0. External factors (such as electromagnetic interference, power line noise, etc.) and experimental conditions (such as auditory and visual stimuli, participant movements, and electrocardiogram signals) can interfere with EEG signals, leading to noise and artifacts. To address these interferences, the data were first filtered using a notch filter centered at 50 Hz, followed by a band-pass filter with a range of 0.5 to 45 Hz. Next, the ensemble empirical mode decomposition-independent component analysis method was applied to enhance artifact suppression, removing signal segments affected by noise, such as blinking, eye movements, and muscle activity [22], making the analysis more reliable. The data were then visually inspected to exclude segments containing noise. Finally, the EEG data were re-referenced. The processed data were then used for subsequent computational analysis.

### 2.5. Entropy Method

#### 2.5.1. Sample Entropy

SaEn is a measure of the complexity of a time series, proposed by Richman and Moorman et al. [23]. SaEn does not consider comparisons with the same data segment (i.e., when i = j) and is not affected by the length of the time series [24], resulting in smaller calculation errors. Given a time series of length N and two parameters, the embedding dimension (m) and the similarity tolerance threshold (r), SaEn are defined as follows:(1)$$\mathrm{SaEn}=lnB^{m}(r)-lnA^{m}(r)$$
where lnBm(r) and lnAm(r) represent the similarity comparison ratios for dimensions m and m + 1, respectively. These ratios are calculated as follows:(2)$$B^{m}(r)=\frac{1}{N-m}\sum_{i=1}^{N-m}B_{i}^{m}(r)$$(3)$$A^{m}(r)=\frac{1}{N-m}\sum_{i=1}^{N-m}A_{i}^{m}(r)$$

In this study, we specified m = 2 and r = 0.15 × D (where SD is the standard deviation of the original time series). Since SaEn does not have specific requirements for the length of the time series, the entire data segment was used for the analysis.

#### 2.5.2. Multiscale Sample Entropy

MSE is an extended method of sample entropy that captures the complexity of a time series across different time scales through a coarse-graining process [25]. The calculation process of MSE began with down-sampling to generate subsequences of varying scales. Each subsequence had a length of N/s, where N represents the length of the original sequence, and s represents the scale factor. Subsequently, SaEn was computed for each scale, and finally, the average was computed to derive the MSE. The formula is expressed as follows:(4)$$y_{j}^{s}=\frac{1}{s}\sum_{i=(j-1)s+1}^{js}x_{i},1\leq j\leq \frac{N}{s}$$

With parameters consistent with SaEn (m = 2 and r = 0.15 × SD), we investigated changes in entropy values across 20 scale factors, denoted as s, ranging from 1 to 20.

#### 2.5.3. Phase Transfer Entropy

TE is an asymmetric quantitative measure based on the framework of information theory. Its principle can be described as the extent to which the known information from the source signal reduces the uncertainty of the future state of the target signal [10,26]. The TE value from the source signal (X) to the target signal (Y) can be expressed as follows:(5)$${\mathrm{TE}}_{X\to Y}=\sum p(y_{t+\delta },y_{t},x_{t})log\frac{p(y_{t+\delta }\;|\;y_{t},x_{t})}{p(y_{t+\delta }\;|\;y_{t})}$$
where δ represents the prediction delay of the time series X and Y.

Time series can be described by amplitude and instantaneous phases [27]. Lobier et al. extended the concept of instantaneous phase to TE, using the Hilbert transform to extract the instantaneous phase as input for the TE function [14]. Studies have shown that if the signal is narrowband, the instantaneous phase of the signal can be correctly defined [28]. Therefore, for the EEG signals, the instantaneous phase was computed after filtering the signals in the δ (0.5–4 Hz), θ (4–8 Hz), α (8–13 Hz), and β (13–30 Hz) frequency bands. For the signal X(t), the analytic signal was defined as S(t)=A(t)exp(iθ(t)), where θ(t) is the instantaneous phase time series, and A(t) is the instantaneous amplitude of the signal X(t) [29]. Thus, the PTE from signal X(t) to Y(t) is defined as follows:(6)$$\begin{array}{ll} PTE_{X\to Y}= & H(\theta_{y}(t),\theta_{y}(t’))+H(\theta_{y}(t’),\theta_{x}(t’))\\ & -H(\theta_{y}(t’))-H(\theta_{y}(t),\theta_{y}(t’),\theta_{x}(t’)) \end{array}$$
where $\theta_{x}(t’)$ and $\theta_{y}(t’)$ represent the past states of time series X(t) and Y(t) at time t’=t−δ, respectively. The probability function is obtained by constructing a histogram [14]. In this experiment, the bin width was described by $3.5\sigma /N^{\frac{1}{3}}$ [30], where N is the number of samples and σ is the standard deviation of the direction variable. The prediction delay was set as ${(N}_{s}\times N_{ch})/N_{\pm }$, where N_s_ and N_ch_ represent the number of samples and the number of channels, respectively, and N_±_ is the number of phase change signs across time and channels [31].

Finally, due to the lack of a meaningful upper bound for PTE and to reduce bias, normalized PTE was used, as described in [32]:(7)$${\mathrm{dPTE}}_{xy}=\frac{PTE_{xy}}{PTE_{xy}+PTE_{yx}}$$

The range of dPTE was from 0 to 1. A value of 0.5 < dPTE < 1 indicated that the direction of information flow was from signal X to signal Y, while 0 < dPTE < 0.5 indicated that the direction of information flow was from signal Y to signal X. When the information flow between signal X and signal Y was balanced, dPTE = 0.5.

### 2.6. Statistical Analysis

In this study, we first assessed the normality of the data. Due to the small sample size, the Shapiro–Wilk test was used to verify whether the data followed a normal distribution. Based on this, independent samples *t*-tests were performed to compare the differences in entropy values and electrode node input–output strength between the ASD group and the TD group. Additionally, to analyze changes in the ASD children before and after the 40 sessions of tDCS intervention, paired samples *t*-tests were conducted to compare differences in entropy values and electrode node input–output strength. For the evaluation of intervention effects, paired samples *t*-tests were performed on the ABC scale scores before and after the intervention using SPSS 26 software. The significance level was set at *p* < 0.05. For data that passed the normality test, further False Discovery Rate (FDR) correction was applied to control for the false positive rate in multiple comparisons. In addition, this study conducted a sensitivity analysis using G*Power software version 3.1.9.7 based on the significance level and calculated the effect size using statistical validity and sample size. The results showed that the effect size for the independent samples *t*-test was 0.826, and the effect size for the paired samples *t*-test was 0.61.

## 3. Results

### 3.1. Comparison of Differences Between the ASD Group and the TD Group

#### 3.1.1. Comparison of EEG Complexity Results Between the Two Groups of Children

Compared to the TD group, the ASD group exhibited generally lower SaEn values across all brain regions, as shown in Figure 1A. Except for the left temporal lobe and right occipital lobe regions, the SaEn values in most other brain regions of the ASD group were significantly lower, including the left frontal lobe (t = −2.8599, *p* = 0.0062), right frontal lobe (t = −3.4931, *p* = 0.001), left central region (t = −3.6878, *p* < 0.001), right central region (t = −3.5484, *p* < 0.001), right temporal lobe (t = −2.6604, *p* = 0.0105), and left occipital lobe (t = −2.5676, *p* = 0.0133).

In terms of MSE, both the ASD and TD groups showed an increasing trend in MSE values with the scale factor. However, compared to the TD group, the ASD group exhibited lower MSE values across all scales in all brain regions, as shown in Figure 1B. By averaging the SaEn across all time scales, we calculated the average MSE value at each electrode location. The results indicated that the average MSE of the ASD group was significantly lower than that of the TD group, with statistically significant differences, as shown in Table 1.

#### 3.1.2. Comparison of EEG Effective Connectivity Results Between the Two Groups of Children

Based on PTE, Figure 2A shows the connections with significant differences between the ASD group and the TD group across four frequency bands. The red arrows indicate that the PTE value in the ASD group was significantly higher than that in the TD group, while the blue arrows indicate that the PTE value in the ASD group was significantly lower than that in the TD group. The thickness of the connecting lines represents the t-values after FDR correction. The results show that the PTE values in the ASD group were significantly higher than those in the TD group in the δ, θ, and α bands, while the PTE value in the β band was significantly lower in the ASD group compared to the TD group. To further analyze the PTE differences between the two groups, we calculated the directional connectivity based on PTE and displayed the dPTE between pairs of channels for the ASD and TD groups, as shown in Figure 2B. The results indicated significant reverse information flow in the θ, α, and β bands between the ASD and TD groups. In the figure, red represents the direction of information flow in the ASD group, while blue represents the direction of information flow in the TD group. The thickness of the lines represents the magnitude of the t-values.

Additionally, to further investigate the differences in PTE connectivity weights between the two groups, we compared the node strength between the groups. Node strength refers to the sum of all connection weights linked to a node. Given that the connections were directional, node strength was divided into in-strength and out-strength. The results of the in-strength comparison are shown in Figure 3A. In the δ band, the ASD group exhibited significantly lower in-strength in the left temporal lobe (*p* = 0.025) and right temporal lobe (*p* = 0.012) compared to the TD group. In the θ band, the ASD group showed significantly higher in-strength in the left frontal lobe (*p* = 0.013), left temporal lobe (*p* = 0.001), left central region (*p* = 0.003), and right temporal lobe (*p* = 0.001) compared to the TD group. In the α band, the ASD group had significantly higher in-strength in the left frontal lobe (*p* = 0.009), right frontal lobe (*p* = 0.008), left temporal lobe (*p* = 0.048), left central region (*p* = 0.001), right central region (*p* = 0.003), left occipital lobe (*p* = 0.007), and right occipital lobe (*p* = 0.004) compared to the TD group. In the β band, the ASD group exhibited significantly lower in-strength in the right central region (*p* = 0.021) compared to the TD group. The results of the out-strength comparison are shown in Figure 3B. In the δ band, the ASD group had significantly lower out-strength in the right temporal lobe (*p* = 0.01) and left occipital lobe (*p* = 0.006) compared to the TD group. In the θ band, the ASD group showed significantly higher out-strength in the right frontal lobe (*p* = 0.002) and right central region (*p* = 0.001) compared to the TD group. In the α band, the ASD group had significantly higher out-strength in the left temporal lobe (*p* = 0.006) and right temporal lobe (*p* = 0.001) compared to the TD group. In the β band, the ASD group exhibited significantly lower out-strength in the left temporal lobe (*p* = 0.001) and right temporal lobe (*p* = 0.007) compared to the TD group.

### 3.2. Comparison of Pre- and Post-tDCS Differences

#### 3.2.1. Differences in Brain Complexity Results

As shown in Figure 4A, after 40 sessions of tDCS intervention, the SaEn values in the true stimulation group exhibited an increasing trend across the entire brain, with significant differences observed in the central region (t = 2.5487, *p* = 0.0271). In the MSE analysis, all scale factors showed an upward trend, with the increase in the left and right temporal lobes being particularly pronounced, as shown in Figure 4B. However, statistical tests based on the average MSE values for each electrode did not reveal significant differences, as detailed in Table 2. Meanwhile, no significant differences were observed in the SaEn and MSE values in the sham stimulation group.

#### 3.2.2. Differences in Effective Connectivity Results

Figure 5A shows the connections with significant differences before and after intervention in the real stimulation group. The blue arrows indicate that the PTE value after intervention was significantly lower than before intervention, and the thickness of the connecting lines represents the t-values after FDR correction. The results show that, after intervention, the PTE values in the δ and θ frequency bands were significantly lower than those before intervention in the real stimulation group. Figure 5B presents the dPTE with significant changes between pairs of channels before and after the intervention. The results indicated that, in the θ and β frequency bands, the direction of information flow changed after the intervention compared to before. Red represents the direction of information flow after the intervention, while blue represents the direction before the intervention. The thickness of the lines represents the magnitude of the t-values. No significant changes in effective connectivity were observed before and after modulation in the sham stimulation group.

The node in-strength and out-strength before and after the intervention are shown in Figure 6. The results of the in-strength comparison indicated that in the α frequency band, the in-strength values after the intervention were significantly lower than before in the left frontal lobe (*p* = 0.027), right frontal lobe (*p* = 0.045), left temporal lobe (*p* = 0.049), left central region (*p* = 0.016), right central region (*p* = 0.047), left occipital lobe (*p* = 0.032), and right occipital lobe (*p* = 0.012). The results of the out-strength comparison indicated that in the α frequency band, the out-strength values after the intervention were significantly lower than before in the left frontal lobe (*p* = 0.047), left temporal lobe (*p* = 0.047), and left central region (*p* = 0.027). There were no significant changes in the node in- and out-strength before and after modulation in the sham stimulation group.

### 3.3. Scale Evaluation Results

In this study, the ABC scale was used to assess the regulatory effect of tDCS on the behavior of ASD children, with evaluations conducted before and after the intervention for both groups of ASD children. The ABC scale scores for the real stimulation group are shown in Figure 7. Statistical analysis revealed that, after the intervention, the real stimulation group showed a decrease in all scores, with significant differences observed in the total score (*p* = 0.023) and the social interaction sub-score (*p* = 0.04). In contrast, the ABC scale scores for the sham stimulation group showed no significant changes.

## 4. Discussion

The primary aim of this study was to assess the effects of tDCS intervention on brain complexity and effective connectivity modulation in children with ASD. Previous studies have used symmetric measurements without considering the direction of connectivity. Therefore, using effective connectivity that handles causal interactions between brain regions to study the direction of information flow between brain areas may reveal more insights into brain function. PTE, developed based on information theory, is a model-free effective connectivity metric that not only reveals the direction of interactions but also encompasses both linear and nonlinear interactions while innovatively integrating the phase information of neural oscillations. This study utilized methods such as SaEn, MSE, and PTE to analyze the differences in neural network complexity and information flow between children with ASD and TD children. It also examined the effects of tDCS intervention on brain activity in children with ASD. The main findings of the study are described below.

### 4.1. Brain Complexity and Effective Connectivity Abnormalities Between Two Groups

In this study, we first found that children with ASD exhibited significantly lower SaEn and MSE values compared to the TD group, indicating reduced brain network complexity in ASD children. SaEn, as a measure of time-series complexity, reflects the unpredictability of brain signals. MSE, on the other hand, reflects the complexity of signals across different time scales and can reveal dynamic changes in brain activity at different levels [33]. Previous studies have also found that neural networks in children with ASD tend to exhibit lower complexity, which is consistent with our findings [6,34]. The reduced complexity may be related to cognitive and social impairments in children with ASD, reflecting a lack of flexibility and adaptability in their brain’s processing of information. The decrease in neural network complexity could stem from either local overconnectivity or insufficient global connectivity, a pattern commonly observed in neurodevelopmental disorders.

In the analysis of effective connectivity, we assessed the information transfer ability between the ASD and TD groups across different frequency bands using PTE. The results revealed that the PTE in the ASD group was significantly higher than that in the TD group in the δ, θ, and α bands, while it was significantly lower in the β band. This suggests that the children with ASD exhibited excessive effective connectivity in the low-frequency bands and insufficient effective connectivity in the high-frequency bands, which may reflect a dysregulation in information processing across different frequency bands. Particularly, in the low-frequency range, excessive connectivity may lead to local over-synchronization within the neural network, thereby affecting the brain’s functional integration. Orekhova et al., by analyzing EEG connectivity in infants with ASD, found significant hyperconnectivity in the frontal-central regions in the α-band. This hyperconnectivity was independent of behavioral state and spectral power differences, suggesting that it may reflect early white matter abnormalities or an imbalance in neural excitation/inhibition [35]. In the high-frequency range, β waves are associated with attention, active task engagement, and motor behavior [36]. Insufficient effective connectivity may impede information flow, thereby affecting the brain’s cognitive and motor functions. Notably, the opposite direction of information flow observed between the ASD and TD groups in certain frequency bands further suggested differences in the information transfer patterns between the two groups. The reversed information flow in the θ, α, and β bands may be related to the difficulties ASD children face in processing social, language, and motor functions.

### 4.2. The Effects of tDCS on Brain Complexity and Effective Connectivity in Children with ASD

This study showed that after tDCS intervention, the SaEn and MSE values of the children with ASD significantly increased, indicating that tDCS can significantly enhance brain complexity. This result suggests that tDCS may improve the complexity of the brain network in children with ASD by modulating the activation patterns of neurons, which could improve their behavioral abilities. Previous studies have suggested that the imbalance between neuronal network excitation/inhibition (E/I) may be a key factor in the pathogenesis of ASD, and brain complexity in children with ASD is closely related to E/I imbalance in neuronal networks [37]. Additionally, some studies have found that severe behavioral deficits in ASD are caused by an increased cellular balance between excitation and inhibition within neural microcircuits [38]. Entropy values are considered an EEG surrogate marker for E/I imbalance [39]. We hypothesize that tDCS intervention may enhance brain complexity by balancing neuronal excitation and inhibition, thereby improving behavioral deficits. Furthermore, in conjunction with the results from the behavioral scales, we observed significant reductions in both the total score of the ABC and the social interaction subscale in children with ASD, which further supports our hypothesis. Therefore, future research should explore in depth the specific mechanisms by which tDCS modulates the E/I balance in neuronal networks and assess whether changes in this balance are associated with behavioral improvements in children with ASD. Furthermore, existing research has shown that children with ASD exhibited localized overactivation or inhibition in certain brain regions, which may lead to imbalanced information processing [40]. As a non-invasive brain stimulation technique, tDCS has been shown to modulate the activity of specific brain regions [20], thereby improving the brain’s functional integration and segregation abilities. The results of tDCS intervention in our study further support this view, suggesting that tDCS may improve the neural function of children with ASD by enhancing brain complexity.

Additionally, after tDCS intervention, the PTE in the δ and θ frequency bands significantly decreased, and the direction of information flow changed in the θ and β frequency bands. This suggested that tDCS intervention had a positive impact on the effective connectivity of children with ASD, particularly in addressing the issues of excessive connectivity in low-frequency bands and insufficient connectivity in high-frequency bands. However, we calculated the Pearson correlation coefficient between the ABC behavioral scale and dPTE, and unfortunately, no significant positive or negative correlation was found between the two. Future research could further explore the relationship between the dynamic changes in dPTE and behavioral improvements by combining multimodal neuroimaging, providing a basis for the development of personalized rehabilitation programs.

### 4.3. Limitations and Future Directions

Although this study provided preliminary evidence on the effects of tDCS intervention in modulating brain complexity and connectivity in children with ASD, several limitations remain. First, the sample size of the study was relatively small, involving only children aged 4 to 7 years. Future studies should expand the sample size and include a broader age range to more comprehensively assess the effects of tDCS intervention. Second, this study did not track the long-term effects of the intervention. Future research should evaluate the long-term effects of tDCS to further validate its durability and stability. Additionally, the stimulation location and electrode polarity of tDCS are important factors influencing the intervention’s outcomes. Future studies should explore the impact of different stimulation protocols on brain function in children with ASD, particularly investigating the effects of stimulating other brain regions to optimize intervention strategies.

## 5. Conclusions

The results of this study indicated that there were significant differences in brain complexity and effective connectivity between the children with ASD and TD children. Furthermore, the tDCS intervention significantly improved the brain network complexity and information transfer patterns in the children with ASD. Our study provides preliminary evidence for the application of tDCS in children with ASD, suggesting that it may be a promising intervention tool. Future research can further explore the optimal stimulation parameters, timing of intervention, and its impact on clinical behavioral characteristics of ASD to advance the application of this technology in the treatment of neurodevelopmental disorders.

## Figures and Tables

**Figure 1 bioengineering-12-00283-f001:**
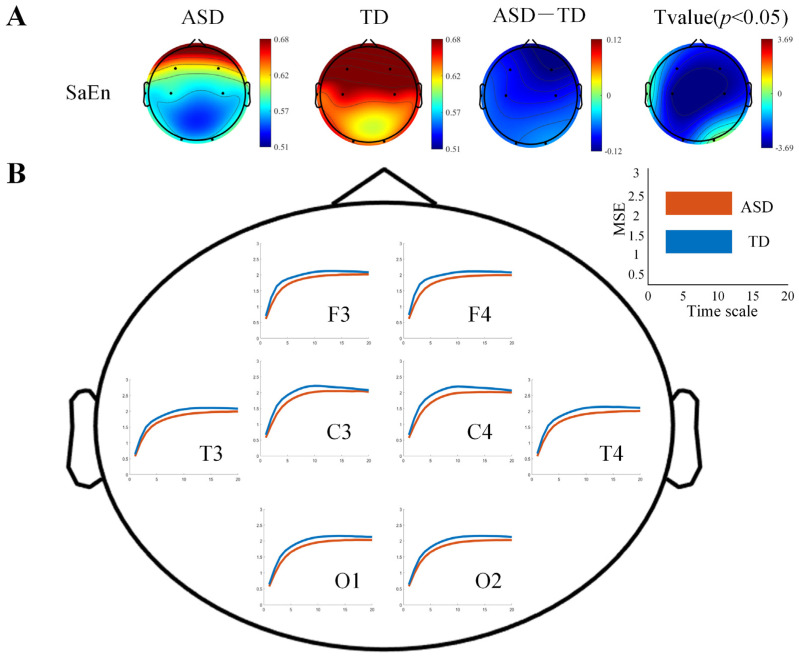
Comparison of EEG complexity differences between the two groups of children. (**A**) The scalp topography of SaEn for the ASD group and the TD group and their significant differences. (**B**) The MSE curves for the ASD group and the TD group. The *x*-axis represents the time scale, and the *y*-axis represents the MSE value. The axis scales of the eight channels are identical, with specific tick marks shown in the upper right corner. The deep orange line represents the ASD group, and the blue line represents the TD group.

**Figure 2 bioengineering-12-00283-f002:**
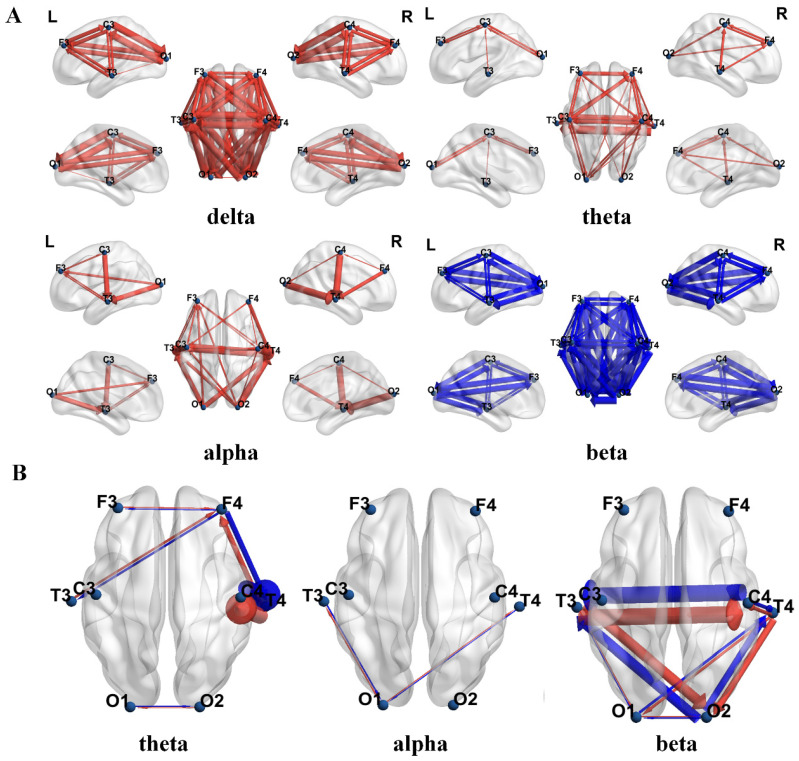
Comparison of differences in EEG effective connectivity results between the two groups of children. The circles represent electrode nodes. (**A**) Statistical comparison of PTE values between the ASD group and the TD group. (**B**) Inter-group differences in dPTE for paired channels between the ASD group and the TD group.

**Figure 3 bioengineering-12-00283-f003:**
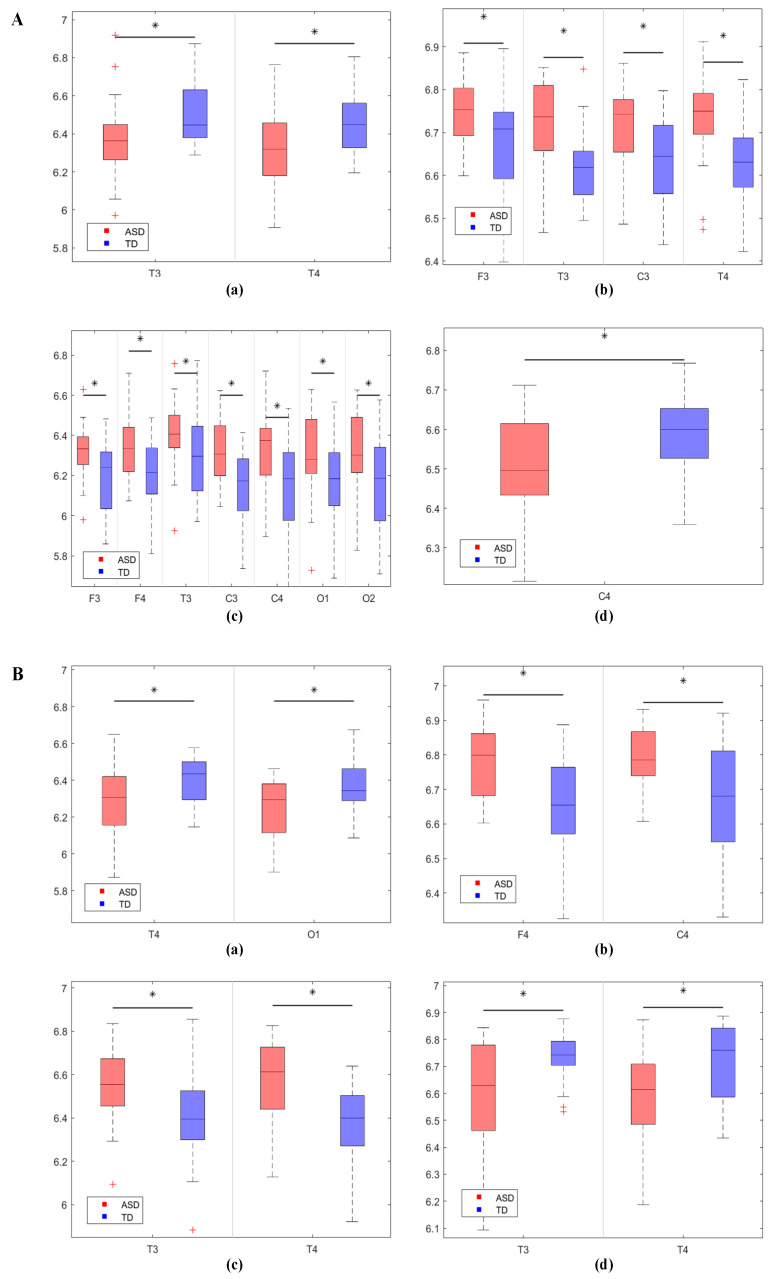
Box plots of the in-strength and out-strength of nodes for the two groups of children, with asterisks indicating significant differences. (**A**) Significant differences in in-strength between the ASD and TD groups, where (**a**–**d**) represent the δ, θ, α, and β frequency bands, respectively. (**B**) Significant differences in out-strength between the ASD and TD groups, with (**a**–**d**) representing the δ, θ, α, and β frequency bands, respectively.

**Figure 4 bioengineering-12-00283-f004:**
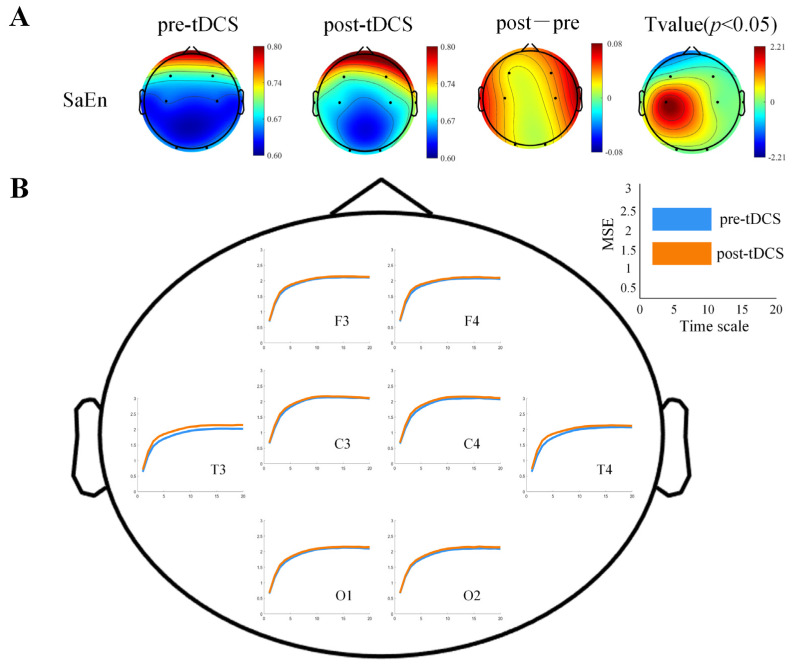
Comparison of EEG complexity differences before and after tDCS modulation in the real stimulation group. (**A**) The scalp topography of SaEn and its significant differences before and after tDCS modulation in the real stimulation group. (**B**) The MSE curves before and after tDCS modulation in the real stimulation group. The *x*-axis represents the time scale, and the *y*-axis represents the MSE value. The orange line represents post-intervention, and the blue line represents pre-intervention.

**Figure 5 bioengineering-12-00283-f005:**
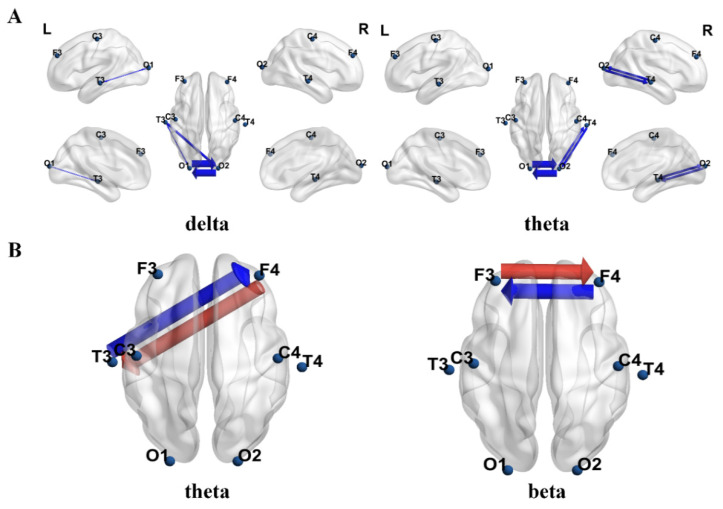
Comparison of the differences in EEG effective connectivity before and after modulation in the true stimulation group. (**A**) Significant differences in PTE values before and after modulation in the true stimulation group. (**B**) Differences in dPTE between pairs of channels before and after modulation in the true stimulation group.

**Figure 6 bioengineering-12-00283-f006:**
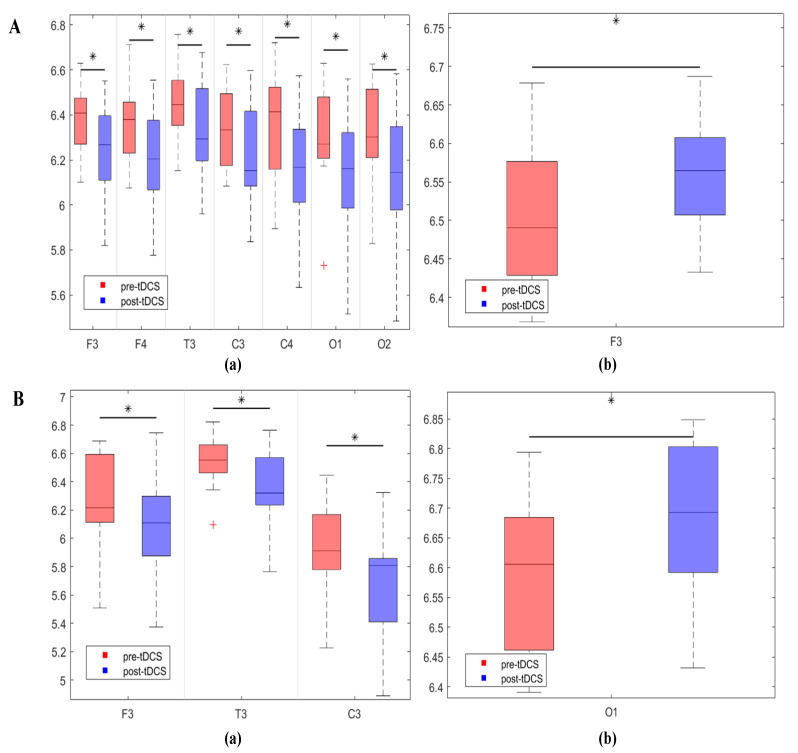
Comparison of the differences in node in-strength and out-strength before and after modulation in the true stimulation group. Asterisks in the figure indicate significant differences. (**A**) Significant differences in in-strength before and after modulation in the true stimulation group, where (**a**) and (**b**) represent the α and β frequency bands, respectively. (**B**) Significant differences in out-strength before and after modulation in the true stimulation group, where (**a**) and (**b**) represent the α and β frequency bands, respectively.

**Figure 7 bioengineering-12-00283-f007:**
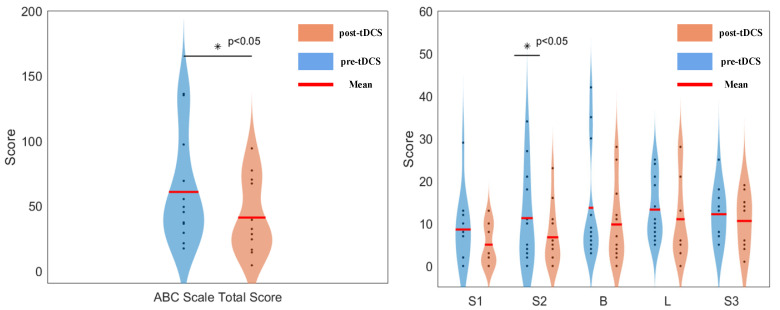
Significant differences in ABC scale scores before and after modulation in the true stimulation group. S1—sensory, S2—social interaction, B—body and object use, L—language, S3—social and self-help.

**Table 1 bioengineering-12-00283-t001:** The average MSE values for all electrodes of two groups.

Electrode	ASD Group	TD Group	MSE *p*-Value
F3	1.7920	1.9258	0.0035 *
F4	1.7814	1.9362	0.0003 *
T3	1.7474	1.8786	0.0293 *
C3	1.8183	1.9660	0.0012 *
C4	1.7906	1.9524	0.0012 *
T4	1.7631	1.9165	0.002 *
O1	1.7810	1.9190	0.0004 *
O2	1.7798	1.9134	0.0033 *

(* indicates a significant difference).

**Table 2 bioengineering-12-00283-t002:** The average MSE values for each electrode before and after modulation in the real stimulation group.

Electrode	pre-tDCS	post-tDCS	MSE *p*-Value
F3	1.8966	1.9310	0.261
F4	1.8812	1.9252	0.354
T3	1.8003	1.9267	0.215
C3	1.9101	1.9475	0.234
C4	1.8820	1.9390	0.151
T4	1.8381	1.9235	0.172
O1	1.8862	1.9253	0.277
O2	1.8740	1.9203	0.252

## Data Availability

The data of this study are available from the corresponding author upon reasonable request.

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
