# Peer review of "Transcranial Direct Current Stimulation Can Modulate Brain Complexity and Connectivity in Children with Autism Spectrum Disorder: Insights from Entropy Analysis"

_bioengineering, 2025, doi:10.3390/bioengineering12030283_

Round 1
Reviewer 1 Report
Comments and Suggestions for Authors
The study explores the effects of transcranial direct current stimulation (tDCS) on brain complexity and effective connectivity in children with autism spectrum disorder (ASD) using entropy-based EEG analysis. Results indicate that tDCS has the potential to enhance brain complexity and modulate effective connectivity in children with ASD, leading to behavioral improvements. These findings suggest that tDCS could be a promising therapeutic approach for ASD intervention. However, further research is needed to refine stimulation protocols and assess its long-term effects.
The topic is relevant, and the methodological approach is highly advanced and well-structured, leveraging entropy measures such as Sample Entropy (SaEn), Multiscale Entropy (MSE), and Phase Transfer Entropy (PTE) to evaluate neural complexity and information transfer. The randomized controlled trial (RCT) design with sham stimulation strengthens the validity of the conclusions; also, the combination of EEG analysis with behavioral measures (Autism Behavior Checklist, ABC) provides a comprehensive evaluation of tDCS effects.
While the study presents valuable findings, there are some points that the authors should clarify.
The study includes only 24 ASD and 24 TD children, with further subdivision into real and sham stimulation groups (12 per group). Such a small sample size limits statistical power and increases the likelihood of false positives or negatives. Furthermore, the authors perform statistical tests but do not provide effect sizes for the reported differences. Given the small sample, effect sizes should be included to assess the practical significance of the results.
The authors claim that ASD children exhibit excessive connectivity in low-frequency bands (δ, θ, α) and reduced connectivity in the β band. However, this interpretation should be linked more explicitly to the broader literature on ASD neurophysiology.
Final observations: the study suggests that tDCS modulates brain complexity and behavior, but does not discuss potential placebo effects. Given the small sample, how was placebo response controlled? Furthermore, ASD is a highly heterogeneous disorder. Did the sample include children with different severity levels? Were cognitive or language abilities controlled?
Author Response
The study explores the effects of transcranial direct current stimulation (tDCS) on brain complexity and effective connectivity in children with autism spectrum disorder (ASD) using entropy-based EEG analysis. Results indicate that tDCS has the potential to enhance brain complexity and modulate effective connectivity in children with ASD, leading to behavioral improvements. These findings suggest that tDCS could be a promising therapeutic approach for ASD intervention. However, further research is needed to refine stimulation protocols and assess its long-term effects.
The topic is relevant, and the methodological approach is highly advanced and well-structured, leveraging entropy measures such as Sample Entropy (SaEn), Multiscale Entropy (MSE), and Phase Transfer Entropy (PTE) to evaluate neural complexity and information transfer. The randomized controlled trial (RCT) design with sham stimulation strengthens the validity of the conclusions; also, the combination of EEG analysis with behavioral measures (Autism Behavior Checklist, ABC) provides a comprehensive evaluation of tDCS effects.
While the study presents valuable findings, there are some points that the authors should clarify.
The study includes only 24 ASD and 24 TD children, with further subdivision into real and sham stimulation groups (12 per group). Such a small sample size limits statistical power and increases the likelihood of false positives or negatives. Furthermore, the authors perform statistical tests but do not provide effect sizes for the reported differences. Given the small sample, effect sizes should be included to assess the practical significance of the results.
A:Thank you for your valuable comments. This study used G*Power software to conduct a sensitivity analysis based on the alpha level. The effect size was calculated using statistical power and sample size. The results showed that the effect size for the independent samples t-test was 0.826, and the effect size for the paired samples t-test was 0.61. We have revised the statistical analysis section in the manuscript and highlighted the changes in red. (Page 6, line 251)
The authors claim that ASD children exhibit excessive connectivity in low-frequency bands (δ, θ, α) and reduced connectivity in the β band. However, this interpretation should be linked more explicitly to the broader literature on ASD neurophysiology.
A:Thank you for your valuable comments. We have made revisions to the discussion section of the manuscript and highlighted the changes in red. (Page 14, line 445)
Final observations: the study suggests that tDCS modulates brain complexity and behavior, but does not discuss potential placebo effects. Given the small sample, how was placebo response controlled? Furthermore, ASD is a highly heterogeneous disorder. Did the sample include children with different severity levels? Were cognitive or language abilities controlled?
A:Thank you for your valuable comments. In the experiment, we set up a sham stimulation group; however, upon observing the results of the sham group, we found no significant differences before and after tDCS intervention. We apologized for not clearly presenting this in section “3.2.2 Differences in effective connectivity results” and have now added it to the original text, marked in red(Page 12, line 376; page 13, line 391). The limitations of the current study included a small sample size and the lack of systematic control for ASD heterogeneity. Future research will increase the sample size to verify the robustness of the results.
Reviewer 2 Report
Comments and Suggestions for Authors
This study investigates the effects of transcranial direct current stimulation (tDCS) on brain complexity and connectivity in children with Autism Spectrum Disorder (ASD) using entropy analysis. While the topic is relevant and the findings are interesting, several major issues need to be addressed before publication. Below are the key concerns and suggestions for improvement:
1. The manuscript does not sufficiently discuss the limitations of prior research on tDCS in ASD. While the paper highlights the effectiveness of tDCS, it does not clearly address how the study overcomes the shortcomings of previous work. The discussion should include an analysis of what previous studies have failed to account for and why information entropy methods provide a novel advantage in this context.
2. The authors do not provide a detailed explanation of the rationale behind their sample size selection. Given that only 24 children with ASD were included, it is unclear whether the study is sufficiently powered to detect meaningful effects. The manuscript should include a formal power analysis or statistical justification for the chosen sample size.
3. The paper states that participants were randomly assigned to the experimental and sham stimulation groups, but it does not specify who conducted the randomization and whether allocation concealment was implemented. Additionally, it is unclear whether participants and evaluators were blinded to group assignments until the final assessment. The methodology section should explicitly describe the randomization and blinding procedures.
4. The study involves a sham stimulation group, but it does not discuss whether participants in this group received any compensation or an opportunity to receive the actual intervention post-study. To ensure ethical compliance, it is important to clarify whether sham stimulation participants were later offered tDCS treatment.
5. The intervention lasted for 40 sessions, which is quite extensive. However, the manuscript does not discuss dropout rates or participant adherence. If no dropouts occurred, the authors should explain how they maintained high adherence. Additionally, the paper should clarify whether all collected data were included in the final analysis.
6. Since nonparametric tests were used for analysis, it is recommended that results be reported using medians and interquartile ranges rather than means and standard deviations. This will provide a more accurate representation of the data.
7. In addition to addressing the above concerns, the discussion section should be expanded to include the following points:
1) The Clinical Relevance of the Findings: While the study provides evidence that tDCS improves brain complexity and connectivity, it is not clear how these changes translate into real-world functional improvements for children with ASD. The discussion should explore the potential clinical implications of these findings.
2) Long-Term Effects and Sustainability of Improvements: The study only examines short-term outcomes. It would be beneficial to discuss whether the observed improvements are likely to persist beyond the intervention period and whether follow-up studies are planned to assess long-term effects.
Author Response
This study investigates the effects of transcranial direct current stimulation (tDCS) on brain complexity and connectivity in children with Autism Spectrum Disorder (ASD) using entropy analysis. While the topic is relevant and the findings are interesting, several major issues need to be addressed before publication. Below are the key concerns and suggestions for improvement:
1.The manuscript does not sufficiently discuss the limitations of prior research on tDCS in ASD. While the paper highlights the effectiveness of tDCS, it does not clearly address how the study overcomes the shortcomings of previous work. The discussion should include an analysis of what previous studies have failed to account for and why information entropy methods provide a novel advantage in this context.
A:Thank you for your valuable comments. The shortcomings of previous tDCS studies have been addressed in the introduction of the article and highlighted in the manuscript(See page 2, line 94; page 3, line 105). The innovative advantages of the entropy method in this study have been revised in the discussion section and marked in red (Page 14, line 413).
2.The authors do not provide a detailed explanation of the rationale behind their sample size selection. Given that only 24 children with ASD were included, it is unclear whether the study is sufficiently powered to detect meaningful effects. The manuscript should include a formal power analysis or statistical justification for the chosen sample size.
A:Thank you for your valuable comments. For the sample size in this study, a priori analysis was conducted using G*Power software based on the significance level. Statistical power (1-β) and effect size were used to calculate the sample size. The results indicated that 21 children with ASD were required. A total of 24 children with ASD were included in this study, which was sufficient to detect the expected effect size.
3.The paper states that participants were randomly assigned to the experimental and sham stimulation groups, but it does not specify who conducted the randomization and whether allocation concealment was implemented. Additionally, it is unclear whether participants and evaluators were blinded to group assignments until the final assessment. The methodology section should explicitly describe the randomization and blinding procedures.
A:Thank you for your valuable comments. In terms of randomization, this study used a lottery method for random group assignment, and the randomization process was carried out by an independent member of the research team. In the methods section, we will add a detailed description of the randomization procedure and highlight it(Page 3, line 139). Regarding blinding, no blinding was implemented during the experiment. Participants were able to perceive whether they received the sham stimulus during the intervention, making it difficult to ensure complete blinding.
4.The study involves a sham stimulation group, but it does not discuss whether participants in this group received any compensation or an opportunity to receive the actual intervention post-study. To ensure ethical compliance, it is important to clarify whether sham stimulation participants were later offered tDCS treatment.
A:Thank you for your valuable comments. According to ethical requirements, we did not mandate or assume the provision of intervention treatment. After the study, we offered participants in the sham stimulation group the option to be prioritized for inclusion in subsequent treatment options, allowing them to choose to receive real tDCS treatment, based on their needs and preferences. All measures were in line with ethical guidelines, ensuring transparency and compliance with participants' informed consent.
5.The intervention lasted for 40 sessions, which is quite extensive. However, the manuscript does not discuss dropout rates or participant adherence. If no dropouts occurred, the authors should explain how they maintained high adherence. Additionally, the paper should clarify whether all collected data were included in the final analysis.
A:Thank you for your valuable comments. The study initially recruited 26 children, of whom 2 withdrew before group assignment for personal reasons. As a result, the remaining 24 children were randomly assigned to the real stimulation group and the sham stimulation group. The experiment was conducted at the Children’s Rehabilitation Center of Ningbo Rehabilitation Hospital. During each intervention, the participants were accompanied by their parents or teachers, ensuring that all 24 participants completed the 40 interventions without dropout. To maintain high compliance, we maintained close communication and cooperation with the parents and teachers to ensure that the participants received the necessary support and encouragement throughout the intervention. All collected data were included in the final analysis. We performed data preprocessing and ensured the exclusion of any potential outliers. We have made revisions in the section "2.1 Subjects " of the article and highlighted the changes in red(Page 3, line 124).
6.Since nonparametric tests were used for analysis, it is recommended that results be reported using medians and interquartile ranges rather than means and standard deviations. This will provide a more accurate representation of the data.
A:Thank you for your valuable comments. We understand that non-parametric tests should be used when the data do not follow a normal distribution. However, in our data analysis, we performed normality tests, and the results indicated that the data follow a normal distribution. Therefore, we chose to use parametric tests and reported the means and standard deviations.
7.In addition to addressing the above concerns, the discussion section should be expanded to include the following points:
1) The Clinical Relevance of the Findings: While the study provides evidence that tDCS improves brain complexity and connectivity, it is not clear how these changes translate into real-world functional improvements for children with ASD. The discussion should explore the potential clinical implications of these findings.
A:Thank you for your valuable comments. We have made revisions in the discussion section of the manuscript and highlighted them in red(Page 15, line 464; page 15, line 490).
2) Long-Term Effects and Sustainability of Improvements: The study only examines short-term outcomes. It would be beneficial to discuss whether the observed improvements are likely to persist beyond the intervention period and whether follow-up studies are planned to assess long-term effects.
A:Thank you for your valuable comments. This study indeed focuses on the evaluation of the short-term effects of the intervention. Currently, the data cannot directly verify the long-term sustainability, which is a key limitation of this phase of the research. However, we believe that the improvements from the intervention may persist for a period of time after its conclusion. To gain deeper insights into the sustainability of these improvements, we are considering conducting a follow-up longitudinal study, where participants will be assessed at different time points to observe whether the intervention effects are maintained.
Reviewer 3 Report
Comments and Suggestions for Authors
The authors tackle an interesting problem: the examination of how brain complexity can be altered with transcranial stimulation in ASD children. 24 ASD are compared to 19 typical development (a.k.a. normal) children. Relatively simple entropy methods are used to analyze the EEG to establish if there are changes in brain connectivity after transcranial stimulation and/or corresponding changes in behavior. The authors produce a well written paper to describe the study and support its results in context with the present literature. Overall, the paper was relatively strong. Some suggestions for improvement include:
- Better explaining how reduced complexity correlates with ASD mechanisms and better literature review of studies that have shown this
- Standard brain map atlases could be used with the authors results to improve the visualization of changes to connectivty
- The authors did not adequately describe the difference between ASD and TD groups before and after transcranial stimulation. It is understandable that more emphasis would be on ASD after treatment. However, it is important to fully write about the control group as well. The authors need to divide sub-sections so that it is easier to compare the 4 groups (ASD-before treatment; ASD-after treatment; TD-before treatment; TD-after treatment).
- The authors could better expound on the behavioral testing assessment in the Methods (briefly summarize) and make a visualization to illustrate correlations with each behavioral sub-test with associations in connectivity to different parts of the brain or, minimally, fold change in connectivity.
- Figure captions need to better describe which of the 4 groups are being examined (ASD-before treatment; ASD-after treatment; TD-before treatment; TD-after treatment). Be consistent with use of colors in the figures to help with reader understanding. This was a bigger issue with behavioral testing.
- Examine the figures to ensure legibility. Some of the fonts and characters are extremely small and hard to visualize.
Author Response
The authors tackle an interesting problem: the examination of how brain complexity can be altered with transcranial stimulation in ASD children. 24 ASD are compared to 19 typical development (a.k.a. normal) children. Relatively simple entropy methods are used to analyze the EEG to establish if there are changes in brain connectivity after transcranial stimulation and/or corresponding changes in behavior. The authors produce a well written paper to describe the study and support its results in context with the present literature. Overall, the paper was relatively strong. Some suggestions for improvement include:
1.Better explaining how reduced complexity correlates with ASD mechanisms and better literature review of studies that have shown this.
A:Thank you for your valuable comments. We have made revisions in the introduction section of the article and highlighted them in the manuscript. (See page 2, line 51; page 2, line 67)
2.Standard brain map atlases could be used with the authors results to improve the visualization of changes to connectivity.
A:Thank you for your valuable comments. In Figures 2 and 5, we have changed the visualization of brain connectivity (Page 8, line 298; page 12, line 378).
3.The authors did not adequately describe the difference between ASD and TD groups before and after transcranial stimulation. It is understandable that more emphasis would be on ASD after treatment. However, it is important to fully write about the control group as well. The authors need to divide sub-sections so that it is easier to compare the 4 groups (ASD-before treatment; ASD-after treatment; TD-before treatment; TD-after treatment).
A:Thank you for your valuable comments. The experimental design of this study focused on comparing the intervention effects within the ASD children group and did not involve any interventions for the TD children. We randomly divided the ASD children into two groups: the active stimulation group and the sham stimulation group. The TD children in this study were only used as a reference standard and did not participate in the intervention. Additionally, since the sham stimulation group did not show significant differences before and after the intervention, the results are not presented in "3.2 Comparison of pre- and post-tDCS differences." The figure title in the original manuscript for section "3.2 Comparison of pre- and post-tDCS differences " may have caused confusion, and we have made revisions in the manuscript, highlighted in red. ( Page 11, line 355; page 11, line 361; page 12, line 379; page 13, line 394; page 14, line 408)
4.The authors could better expound on the behavioral testing assessment in the Methods (briefly summarize) and make a visualization to illustrate correlations with each behavioral sub-test with associations in connectivity to different parts of the brain or, minimally, fold change in connectivity.
A:Thank you for your valuable comments. The summary of the behavioral testing evaluation has been revised in the article section "2.3 Data Acquisition" and highlighted in red (Page 4, line 166). Additionally, we did calculate the pearson correlation coefficients between the ABC behavior scale evaluation and connectivity in different regions, but unfortunately, there was no significant positive or negative correlation, so the results were not presented in the experimental findings.
5.Figure captions need to better describe which of the 4 groups are being examined (ASD-before treatment; ASD-after treatment; TD-before treatment; TD-after treatment). Be consistent with use of colors in the figures to help with reader understanding. This was a bigger issue with behavioral testing.
A:Thank you for your valuable comments. It should be noted that the study first compared the differences between children with ASD and TD children, and then conducted subgroup interventions for the ASD group, with no tDCS intervention for the TD children. Additionally, in "3.1 Comparison of differences between the ASD group and the TD group ", red represents the ASD group results, while blue represents the TD group results. In "3.2 Comparison of pre- and post-tDCS differences",cool colors (blue) mostly represent pre-intervention, while warm colors (red, orange) mostly represent post-intervention.
6.Examine the figures to ensure legibility. Some of the fonts and characters are extremely small and hard to visualize.
A:Thank you for your valuable comments. We have now made changes to the fonts and characters in the figures.
Round 2
Reviewer 2 Report
Comments and Suggestions for Authors
The authors have addressed well all concerns I raised in a previous review. I think that this revised manuscript is ready for publication.